# A Sustainable Two-Echelon Logistics Model with Shipment Consolidation

Noha A. Mostafa [1,2,*] and Omar Eldebaiky [1]

[1]  Mechanical Engineering Department, The British University in Egypt, El Sherouk City 11837, Egypt
[2]  Industrial Engineering Department, Zagazig University, Zagazig 44519, Egypt
*   Correspondence: noha.mostafa@bue.edu.eg or namostafa@eng.zu.edu.eg

**Abstract:** *Background*: Shipment consolidation is a concept in logistics management in which two or more shipments are transported by using the same vehicle with the aim of using less resources. *Methods*: The objective of this manuscript is to study shipment consolidation and assess its impact on cost environment, to achieve this, a mathematical model was developed to optimize shipment consolidation while reducing the emissions and minimizing the costs. *Results*: A case study from major dairy products manufacturers in Egypt was used to validate the model and evaluate the outcomes. A comparison was made between two transportation models, with and without consolidation. Results show that shipment consolidation reduced the total costs by 40% in addition to consuming less fuel, and consequently producing less emissions. *Conclusions*: These findings emphasize the importance of shipment consolidation and how it can be used to achieve more sustainability in logistics management.

**Keywords:** shipment consolidation; transportation model; sustainability; mid-income countries; dairy supply chain

## 1. Introduction

In the modern business environments, logistics are not only important in manufacturing or goods-based industries, but also for service-based industries such as shipping and delivery services [1]. Furthermore, well implemented logistics is directly linked to enhancing business performance and increasing market share. Consequently, logistics management is considered as one of the most important strategic keys in successful businesses, and if used properly it can be a huge competitive advantage as shown in Figure 1. The basic concept of logistics is to improve the efficiency and effectiveness of several operational activities, such as transportation, warehousing and storage, order processing, material handling, and other information management concerning any related data from the origin point to end user. These include either nationally shipped products or internationally shipped products. Integrating these techniques achieves accurately timed and cost-efficient deliveries that meet the requirements of the contract or the business plan [2].

Logistics has been facing increasing challenges, especially in the recent years with high uncertainties due to crises such as the COVID-19 pandemic, the Russian–Ukrainian war, natural calamities, and increasing global orientation towards sustainable business practices [3]. That is why traditional methods of logistics management are no longer valid to face such challenges. Shipment consolidation is considered one of the most important modern techniques in logistics that many shippers now consider indispensable, due to its huge benefits. Shipment consolidation is combining two or more less than container load (LCL) from different shippers into single full container load shipments (FCL). As soon as the full container shipment is delivered to the desired destination, the shipments are disassembled into LCL again to be sent to its final customer. This method is very convenient as it offers better rates to the shippers than using the LCL. Moreover, shipment

consolidation is not only suitable for shippers that have small shipments or few pallets but is also suitable for shipments from different locations and different suppliers to be collected in one shipment to avoid high rates. Consolidation helps small business owners to deliver their goods as this convenient solution offers them affordable prices [4]. Shipment consolidation plays an important role during crises and pandemics, as in these situations there are certain procedures and measures to be taken such as limiting cargo shipping and transportations between countries. Therefore, shipment consolidation was a supportive option during the COVID-19 pandemic, and it helped a lot of countries to overcome the crisis [5]. Shipment consolidation is cost-efficient since the shippers must pay for the full container even if the used space is less than half of the entire space. Thus, by combining multiple LCL shipments into one FCL shipment, better value for money will be achieved [6].

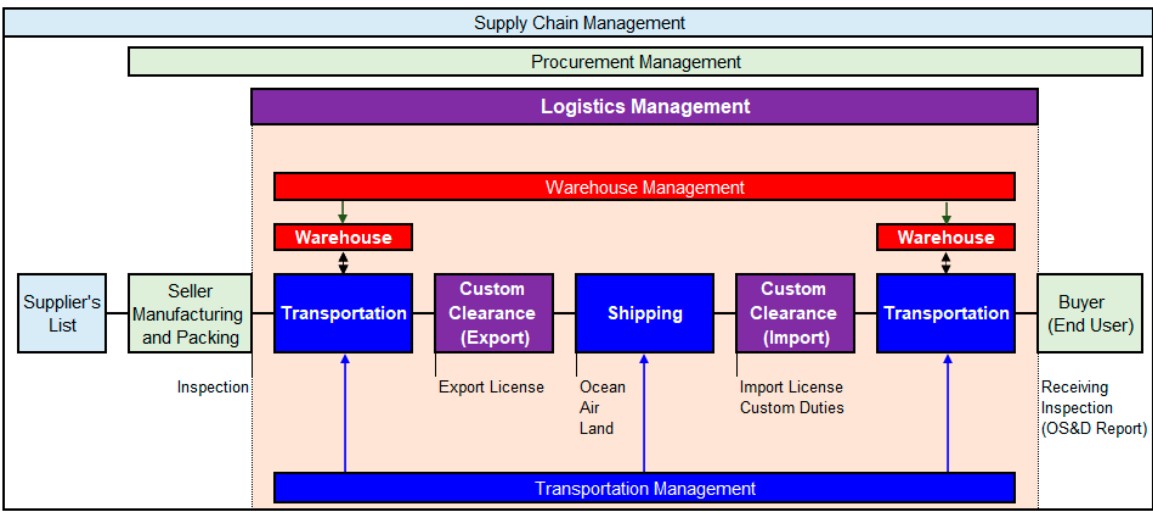

**Figure 1.** Scope of logistics management.

Traditionally, shipment consolidation was all about cost reduction purposes; however, consolidation can also contribute to greening the supply chain. Green Supply Chain Management (GSCM) can be defined as "integrating environmental thinking into supply chain management, including product design, material sourcing and selection, manufacturing processes, delivery of the final product to the consumers as well as end-of-life management of the product after its useful life" [7]. To achieve this in logistics management, several functions can be addressed in addition to consolidation, such as purchasing, inbound logistics, outbound logistics, after-sale service and product returns, recycling, re-manufacturing, and centralised distribution [8].

Moreover, consolidation reduces the damage risk, as fewer touchpoints are offered by the consolidation model reducing the on-again, off-again handling of shipments. Comparatively, the normal freight shipping models contain many stops and multiple touchpoints which increase the chances of damaging the products. One of these models is the hub and spoke distribution method, which mainly allows delivery partners to arrange their daily delivery routes nearby a hub centre, then after finishing all the deliveries in a particular zone relocate to another hub centre for any additional or on-demand deliveries. This can also help against demand volatility [9].

However, there are some challenges that face shipment consolidation that might affect this process. The first problem is securing a carrier, due to the extra complications that come with this process. Even if a good carrier that can transport consolidated shipments was found successfully, there is still the problem of not being charged correctly for only the exact space used. This problem could be solved by dealing with the right carriers with good reputations, or some carriers offer access to their network. This offers a great help to the shippers, as dense networks make it easier to consolidate shipments either nationally

or overseas [10]. Secondly, shipment consolidation takes a lot of time and organising to be planned correctly since there are many factors that should be taken into consideration such as dimensions, estimating the price, timing, and other factors to make sure that the shipment will be delivered safely and on time [11]. Finally, there is the problem that comes when combining the LCL into an FCL as every shipment has certain requirements, such as temperature, humidity, and other factors. For example, if the container contains fast moving consumer goods (FMCG) it cannot be shipped in the same container with electronics, as FMCG have shorter expiry dates, while electronics take more time to be processed due to different customs and procedures, which makes is it hard to keep up with the time limit for the FMCG [12].

The scope of this research paper is the optimisation and sustainability of shipment consolidation in logistics management. Works that considered shipment consolidation were reviewed to understand the analytical models that reduce the carbon footprint, cost, and time in an efficient way that satisfies the requirements of the end user. The main contribution of this paper is to explore how shipment consolidation can achieve improvements in the supply chain, and to develop a model that optimises shipment consolidation while reducing the environmental damage, minimising the costs, and enhancing the process.

## 2. Literature Review

The scope of most of the works on shipment consolidation was to develop new models to control the main factors affecting the consolidation process. These factors include the pickup of the shipments, the delivery method and vehicle routing, cost analysis, and the environmental impact of the model [13]. Each paper has discussed a different model to optimise the best process for shipment consolidation to choose an efficient model to be applied to get the best out of this process. Additionally, a new research direction is to discuss more sustainable options in delivering products and combine them with the shipment consolidation methods to propose a model that is efficient and sustainable. Four shipment consolidation models will be discussed to understand the difference between them and how each model has handled the consolidation method itself. Finally, works that addressed sustainable models are reviewed.

### 2.1. Multi-Product Pickup and Delivery with Location-Routing and Direct Shipment vs. Shipment Consolidation

Currently, most supply chains depend on third party logistics providers (3PL) to outsource their warehousing and logistics operations to improve the efficiency of the supply chain and focus on the production operations. Such activities have been adopted successfully in many companies across multiple industry sectors, such as Wal-Mart [14], Bosch [15], Goodyear [16], and Toyota [11]. 3PL providers integrate the logistics services, warehousing, and operations aiming to meet contractors or end-users needs, such as transportation and pickup services for materials and products. This can be achieved by consolidating shipments that come from different suppliers then store all the shipments for some time and then initiating the delivery process distributing the shipments or products with a fleet of delivery vehicles. This leads to developing a distribution system with low cost and time efficiency, but due to the complexity of arrangements and coordination, optimisation models must be designed to arrange and control the distribution network. The main concern of such models is to specify the vehicle routing, determine the locations of distribution centres, and solve delivery problems that might face the suppliers or the customers.

The main research problems discussed in the literature are the vehicle routing problem (VRP), the location routing problem (LRP), and the vehicle routing problem with cross-docking (VRPCD). The previous literature models mentioned in this paper introduced a multicommodity LRP and solved it with a branch and cut algorithm. In [17], a hybrid heuristic incorporating simulated annealing and artificial algae algorithm to solve the location routing problem with two-dimensional loading constraints. In [11], the problem of integrating cross-docking and vehicle routing was studied and solved through a mathemat-

ical model to find the optimal number of vehicles with main objective to reduce the overall costs. In this model, products were collected by a fleet of vehicles to a distribution centre before delivering it to the customers. Then, in the distribution center the sorting process started, where the goods were to be delivered according to the destinations in a time-saving manner with specific routes timed perfectly to reach the customer quickly. In [18], a Tabu search algorithm was used to improve the solution methods developed by the previous authors. These improvements in the algorithms reached a range of 10% to 36% in some cases. In [19], the objective was to optimise the timing of cross-docking operations for food or fast moving goods to be delivered on time and to reduce the total costs of the system. The system costs consisted of the costs holding of the inventory, the penalty of late or early deliveries, and the delivery costs.

Moreover, most of the previous studies considered the design of single distribution centre in a fixed location, while in [20], a model with two routing types and one distribution centre was proposed. The first type of routing was for a transportation process that was initiated at the cross-dock; after that, it reached a subset of suppliers. The second type started after passing by a subset of suppliers without stopping at the cross-docks. In [21], the VRPCD was discussed relative to different routing for vehicle fleet and scheduling of trucks routes in a multi-door cross-dock system. An estimated sweep-based model was developed to consider several constraints simulating the sweep algorithm. The model was responsible for nodes assigned to vehicles to reduce the search and enhance branch. This model was validated by solving numerical examples for more than fifty transportation requests and different ten vehicle fleets, and the results displayed a reasonable running time. In [22], a mixed integer programming model was used to control the outbound and inbound scheduling of trucks in a cross-docking system. A hybrid algorithm that combines particle swarm with simulated annealing was used to help in solving complex problems in very short time. In [23], the problem of scheduling cross-docking and vehicle routes was addressed for a three-echelons supply chain network. The objective was to reduce late deliveries and delivery costs.

Last mile distribution is one of the most important research topics in supply chain and logistics management. The best example of this process is the last transportation process of the products from supply chains to its last delivery points such as retail stores. Last mile delivery is usually the routing of a fleet of vehicles to stop at a set of delivery destinations, using less than truckload (LTL) or truckload (TL). Various techniques were used throughout the literature to overcome the problems of last mile deliveries and optimise the whole process, such as solving the vehicle routing problem with split deliveries (VRPSD). Although the concept of split deliveries has many benefits in terms of cost, it does not take full advantage of using multiple vehicles to deliver multiple shipments to customers on the same day.

### 2.2. Sustainable Models

A great interest has been into achieving sustainability and greener transportation while creating an efficient system in terms of cost and time. This will inspire the business owners to act and reduce the damage that their companies are responsible for while also making profits to make it a win–win situation. Examining the demand and sustainability of critical metals has focused on light-duty vehicles. Heavy-duty vehicles have often been excluded from the research scope due to their smaller vehicle stock and slower pace of electrification [24]. In 2017, Tesla announced the production of electrical semi-truck with an estimated production start in 2019, but due to the pandemic there was a delay in the production plan. In 2022, Tesla announced the delivery of the first batches to some large companies including Pepsi, Amazon, and Walmart. These semi-trucks raised the level of expectations as it will have a range of 800 km, with a Tesla Mega charger giving 640 km of charge in 30 min. The semi-truck can also use on-site 150 kW charging, taking six to eight hours [25]. There is no doubt, that the electrical heavy-duty vehicles will be considered a game changer in reducing emissions and carbon offsetting. Unfortunately, there are not

enough studies to give an exact number or percentage for emissions compared to diesel trucks as the Tesla vehicles are still new to the market, but certainly this will contribute toward more sustainable delivery model [26].

Another model is the parcel delivery model that identifies the different parameters in the delivery model while considering many combinations of traditional operators (e.g., trucks and vans that use fossil fuel) and green operators (e.g., electric or hybrid vehicles, bikes, and cargo bikes), investigating their business models and behaviours from a managerial perspective [27]. The aim of this model is to form an operational point of view on how to mix low-emission and traditional logistics, especially in urban areas. In [27], a Monte Carlo-based simulation optimisation framework was developed for analysing mixed-fleet board policies related to managing freight delivery in urban areas, clarifying their cost mix (economic and environmental). In [28], the impact of shipment consolidation on home delivery in the retail industry was assessed; the results confirmed the positive role of shipment consolidation on delivery time, total cost, and fuel consumption.

Recent works on the role of logistics in Industry 4.0 have recognised major challenges including cost reduction and resource management [29], which is why shipment consolidation can play a role in enabling Industry 4.0 adoption in logistics management. Shipment consolidation can be seen as an agile method as it improves collaboration in the supply chain and promotes trust among several players [30]. According to [31], shipment consolidation can be beneficial for upstream suppliers and also can achieve Pareto improvement of both economic and environmental sustainability.

### 2.3. Research Questions and Contribution

Most of the discussed models proved that shipment consolidation is a good option in terms of enhancing and optimising logistics efficiency. However, there are not enough works on industrial applications, especially on the food industry, which has the critical perishability factor. In this work, the scope is to develop a new analytical model aiming at reducing costs, saving time, and making the transportation process as environmentally friendly as possible.

This research paper discusses optimising shipment consolidation and categorisation in logistics management and with considering sustainability. The concept of "mid-route shipment consolidation" is discussed to show its benefits and implementation. To do so, two research questions were developed to drive this research:

*RQ1:* How can all the possible vehicle capacity be utilised in last mile distribution in order to reduce the number of dispatched vehicles, and the on-road time of the vehicles?
*RQ2:* How can the emissions and pollutants produced by the vehicles be reduced?

### 3. Problem Statement and Mathematical Model

The paper discusses the problem of decision making in a distribution network consisting of customers, suppliers, and distribution centres. A problem statement and network description will be given in addition to the assumptions to be used for the model.

### 3.1. Problem Statement and Network Description

Suppliers provide the goods to be delivered to customers by two ways, either through direct shipping to the customers or consolidating the shipments in a distribution centre then delivering it to the customers. The objective of the model is to make a decision regarding the delivery mode to reduce costs and minimise the delivery time, both in the transportation process of the products and in the consolidation of shipments in the distribution centre. The significance of the problem appears significant when there are different delivery routes to each subset of customers to satisfy the needs of the market that is usually dealing with multiple products as shown in Figure 2. In the case that a special group of customers need a single product, the process changes into single product delivery plan with different delivery and pick up routes as shown in Figure 3.

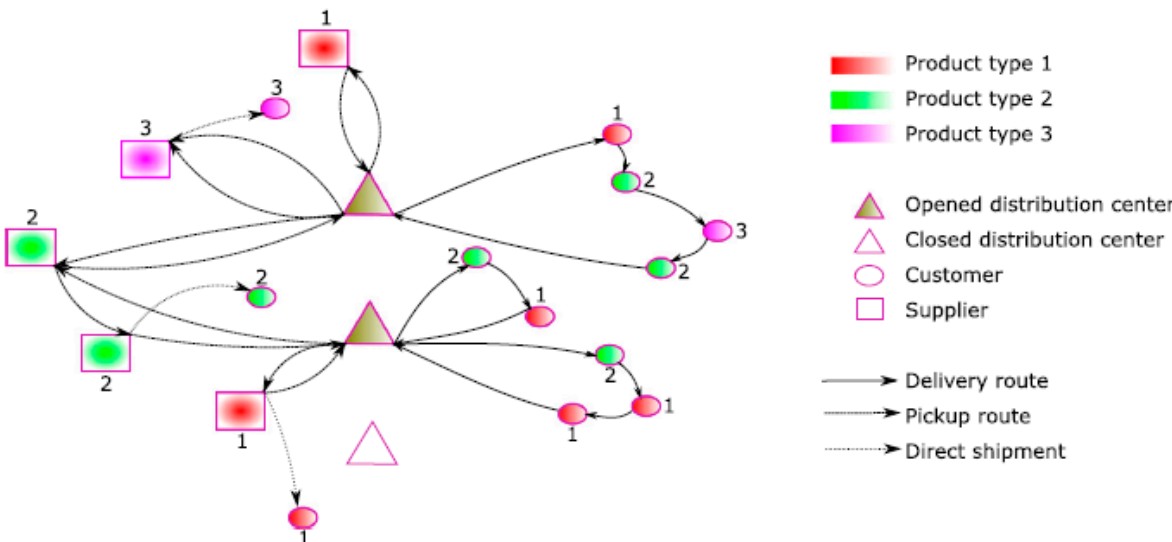

**Figure 2.** Multi-product delivery network (Azizi and Hu).

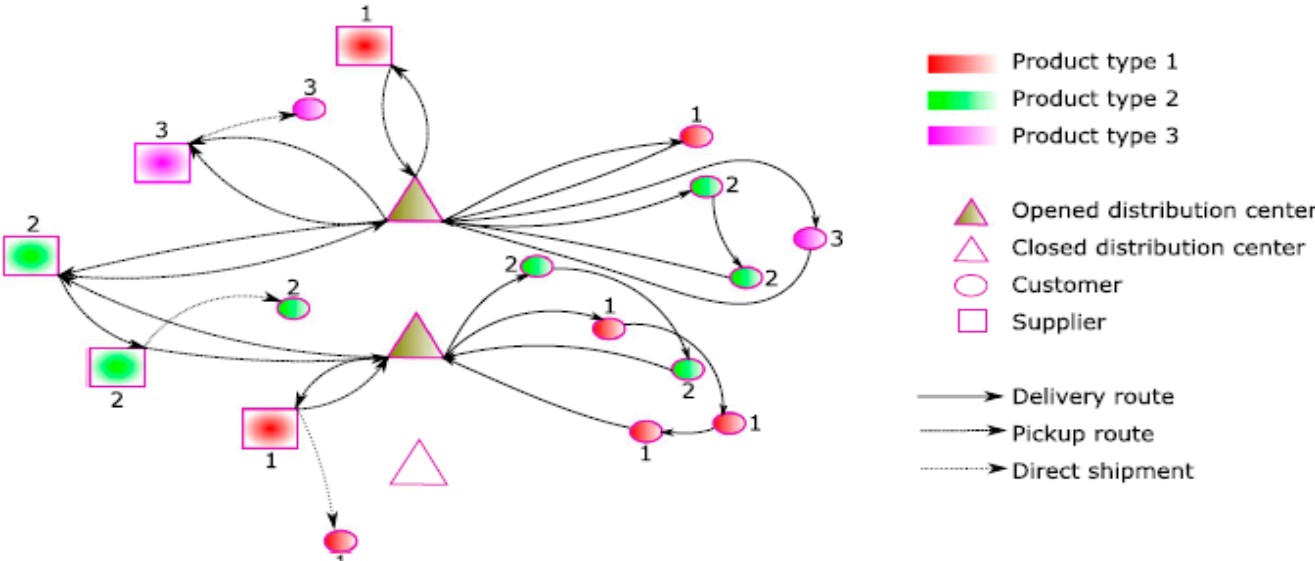

**Figure 3.** Single-product delivery network (Azizi and Hu).

*3.2. Model Assumptions*

The following assumptions will be considered while modelling the problem:

- Each node must belong to one of the following sets: distribution centre node, pickup node (Supplier), or delivery node (End user) [11].
- All the requests for the delivery must be satisfied by either direct shipments or transportation from distribution centres.
- There is a limit on the capacity of the products for each supplier.
- All distribution centres must be provided by vehicles to be able to perform any transportation processes.
- For the delivery and pick up, a fleet of homogeneous vehicles must perform all the delivery processes.
- After finishing each delivery, all the vehicles that are responsible for either pick up or delivery must return to the distribution centre.
- Vehicles can visit pick up nodes more than one time, but delivery nodes are not allowed to be visited more than one time.

The proposal of the mid-route shipment consolidation concept depends on the idea of the synchronisation between the delivery of the product and the availability of the vehicles, as these actions takes place simultaneously. A focus group study was made with the carrier companies to see their opinion about this concept in real-life applications. The feedback described this method as a potential way to reduce costs and to improve vehicle utilisation. On the other hand, it might just need contract adjustments and some extra resources to be implemented. Mid-route shipment consolidation sheds light on the possibility of simple consolidation that can take place by exchanging shipments between vehicles.

In this paper, a transportation model is formulated and then discussed based on a comparison between two scenarios; the first model is a proposed transportation model that emphasises the importance of consolidation by adding multiple distribution centres (DCs) or warehouses between the nodes of the supply (plants) and the demand (hypermarkets). Consolidation takes places at the distribution centres so that the fleet will not go all the way from the factory to the retail store with less than the full truck capacity, as the distribution centres will allow the utilisation of most of the truck capacity, this will also divide the route of the fleet into two phases. The first phase will be from the plant to the distribution centre, while the second phase will be from the distribution centres to the hypermarkets. This consolidated transportation model should save some of the transportation costs, in addition to saving the environment from the pollutants that the fleet will produce if the number of trucks is not reduced. Moreover, the distribution centres will offer extra storage space for the hypermarkets, supplying the needed demand all while eliminating the wasted time and the delay that takes place when ordering from the plant directly. The second model is the current model used in a real dairy supply chain that uses direct shipment delivery from the plants to the hypermarkets. Comparison between the consolidated model and the non-consolidated model is performed to show if there is a significant reduction in cost and emissions between the two models. Dairy products were selected as they are essential and the demand on them is increasing [32].

### 3.3. Linear Programming Transportation Model

The objective function of the model minimises the cost by reducing the handling of the product from the shipper (Plants) to the consolidation points, which are the distribution centres offering the fewest amount of touch points. Additionally, the speed to the market as the strategy of consolidation models depend on delivering the shipments on time, which means that no late deliveries are accepted, since fast moving consumer goods (FMCG) are addressed. In general, there are two types of transportation models, classified into balanced problems and unbalanced problems. The balanced problems are when the demand does not exceed the supply while the unbalanced problems are when the demand exceeds the supply. Since this model deals with hypermarkets as the final destinations, it will be more realistic to deal with balanced transportation problem model to avoid any shortage in the products, as this will cause losses for the hypermarkets. Hence, the model seeks achieving the maximum utilisation of the truck available capacity and providing a faster transportation process. The transportation model can be formulated as the following linear programming model with $m$ origins nodes and $n$ destinations nodes.

$$\textbf{Minimise } \sum_{i=1}^{m}\sum_{j=1}^{n} c_{ij}x_{ij} \tag{1}$$

$$\sum_{j=1}^{n} x_{ij} \leq a_i \qquad i = 1, 2, \ldots\ldots\ldots\ldots m \tag{2}$$

$$\sum_{i=1}^{m} x_{ij} \geq b_j \qquad j = 1, 2, \ldots\ldots\ldots\ldots n \tag{3}$$

$$\sum_{j=1}^{n} x_{ij} + \sum_{i=1}^{m} x_{ij} \leq a_i + b_j \tag{4}$$

$$x_{ij} \geq 0 \tag{5}$$

The objective is to obtain the number of units that will be shipped from origins to destinations taking into account that transportation cost should be minimised while satisfying all the constraints mentioned above. Where $x_{ij}$ is the number of units of shipped from the origin (plant) $i$ to destination (demand) $j$, and $c_{ij}$ is the delivery cost of one unit from the origin $i$ to the destination $j$ expressed in cost per kilometer multiplied by the number of trucks, which is specified as the total amount of units divided by each truck capacity divided by the total number of products. Moreover, $a_i$ is the number of units supplied from the plant (origin) and $b_j$ is the number of units at the destination demand locations.

### 3.4. Distribution Centre Locations

In the real case study, there are no actual distribution centres or warehouses. Nevertheless, the locations of the three warehouses cannot be assumed randomly. The centre of gravity method was used to determine the location of the distribution centres according to the quantity of the two products that will be supplied by the two plants. Table 1 gives the demand and location of each distribution centre with reference point in the middle between the two plants. Using these data, the exact coordinates are calculated and the locations are located on Google maps as shown in Figure 4.

**Table 1.** Demand and location data for each distribution centre.

| DC | | Plant 1 | Plant 2 |
|---|---|---|---|
| DC 1 | Demanded quantity per week | 4500 cartons | 1500 cartons |
| | Relative location | 65 Km N. West | 63.37 Km S. East |
| DC 2 | Demanded quantity per week | 1000 cartons | 2000 cartons |
| | Relative location | 65 Km N. West | 63.37 Km S. East |
| DC 3 | Demanded quantity per week | 1500 cartons | 1500 cartons |
| | Relative location | 65 Km N. West | 63.37 Km S. East |

For DC 1:

X-coordinate: $\dfrac{4500\,(-65 \times \cos 45) + 1500\,(63.75 \times \cos 45)}{6000} = -23.33\ Km$ (West)

Y-coordinate: $\dfrac{4500\,(65 \times \sin 45) + 1500\,(-63.75 \times \sin 45)}{6000} = 23.33\ Km$ (North)

For DC 2:

For X-coordinate: $\dfrac{1000\,(-65 \times \cos 45) + 2000\,(63.75 \times \cos 45)}{3000} = 14.731\ Km$ (East)

For Y-coordinate: $\dfrac{1000\,(65 \times \sin 45) + 2000\,(-63.75 \times \sin 45)}{3000} = -14.731\ Km$ (South)

For DC 3:

For X-coordinate: $\dfrac{1500\,(-65 \times \cos 45) + 1500\,(63.75 \times \cos 45)}{3000} = -0.4419\ Km$ (West)

For Y-coordinate: $\dfrac{1500\,(65 \times \sin 45) + 1500\,(-63.75 \times \sin 45)}{3000} = 0.4419\ Km$ (North)

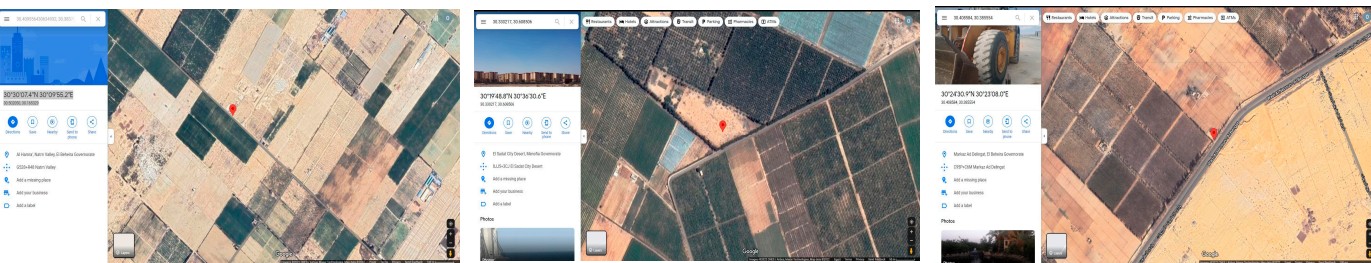

**Figure 4.** Locations of the three Distribution Centres.

*3.5. Cost Analysis*

After determining the locations of the distribution centres the cost of the transportation between the nodes could be easily calculated. However, to estimate the total cost, the number of required trucks should be determined, based on the trucks' capacities. Assuming a fleet of homogeneous tail lift trucks (shown in Figure 5), the number of required trucks is calculated as the total number of cartons to be shipped divided by the capacity of each truck. The following data were collected from the plant. The capacity of each truck is approximately 500 cartons of milk. The average fuel consumption of the tail lift truck is 11 litres per 100 km, which means that the average fuel consumption of the truck equals 0.11 litre per km. Additionally, this kind of truck operates on diesel fuel where the diesel price in Egypt as of May 2022 is EGP 6.750 per litre. Therefore, the price of fuel per km: 0.11 × 6.750 = 0.7425 EGP. The total transportation cost per unit is calculated by using Equation (6). As the truck route is considered a round trip, the shipper charges double the distance because the truck returns empty.

$$Total\ per\ unit\ transportation\ cost = \frac{2 \times Distance\ in\ Km \times Price\ of\ fuel\ per\ Km \times number\ of\ trucks}{number\ of\ units} \tag{6}$$

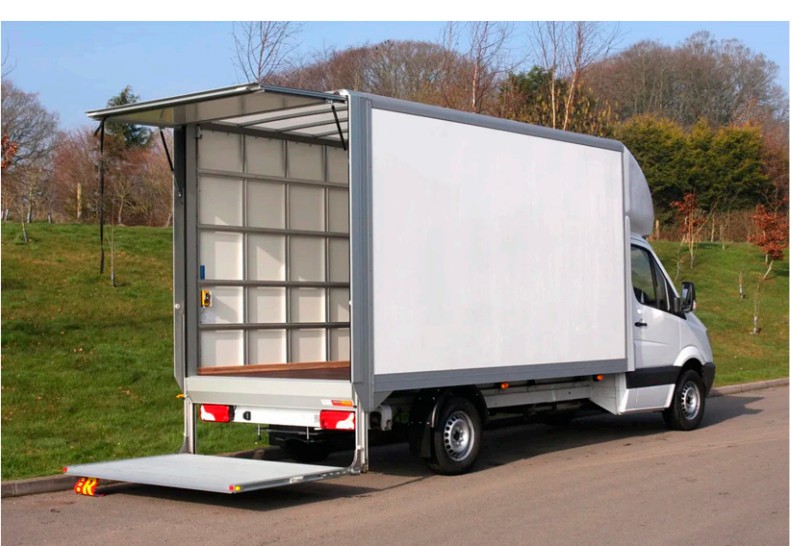

**Figure 5.** A tail lift truck used for transportation of milk cartons.

## 4. Case Study and Results

In this case study, a two-phase transportation model is proposed for a dairy products supply chain. The first phase is between two different large dairy plants (origins) that supply milk cartons to three distribution centres or warehouses (destinations), which will be located between the two suppliers according to centre of gravity method to minimise the distances between the nodes as much as possible. The second phase will be the delivery of the milk cartons from the distribution centres to seven different hypermarket branches located in Cairo. As shown in the model, the supply should always be greater than or equal to the demand to avoid shortage in stock at the hypermarket and to satisfy the condition of balanced transfer problem that was assumed in the model.

The distances between all the nodes of the model are specified after locating the distribution centres that will be elaborated in the site analysis. Data on the capacity of the trucks, the weekly production, and the forecasted demand of milk cartons were collected from the two plants. In comparison, the non-consolidated model will be a delivery model between the two plants and the end destinations without using the distribution centres. The result of the two-phase model will be compared to the non-consolidated model to observe the difference between the two delivery models.

### 4.1. The Consolidated Model

The consolidation model is the model that contains the distribution centers between the nodes of the origin and the end destination, it is divided into two phases (echelons). The first echelon is between the plants and the distribution centres. While the second phase is between the distribution centres and seven branches of a large hypermarket chains as shown in Figure 6 and the distances between the nodes are given in Tables 2 and 3. Figures 7 and 8 show the first and second phases, respectively. Each distribution centre is replenished from both plants; however, each demand point is replenished by a single distribution centre (based on the distance). Tolls are estimated to be EGP 200 per truck per round trip. The reason for breaking this model into two phases is to solve the objective function with fewer constraints and to be simpler while solving. Moreover, the result from the two phases will be combined to be compared with the non-consolidated model.

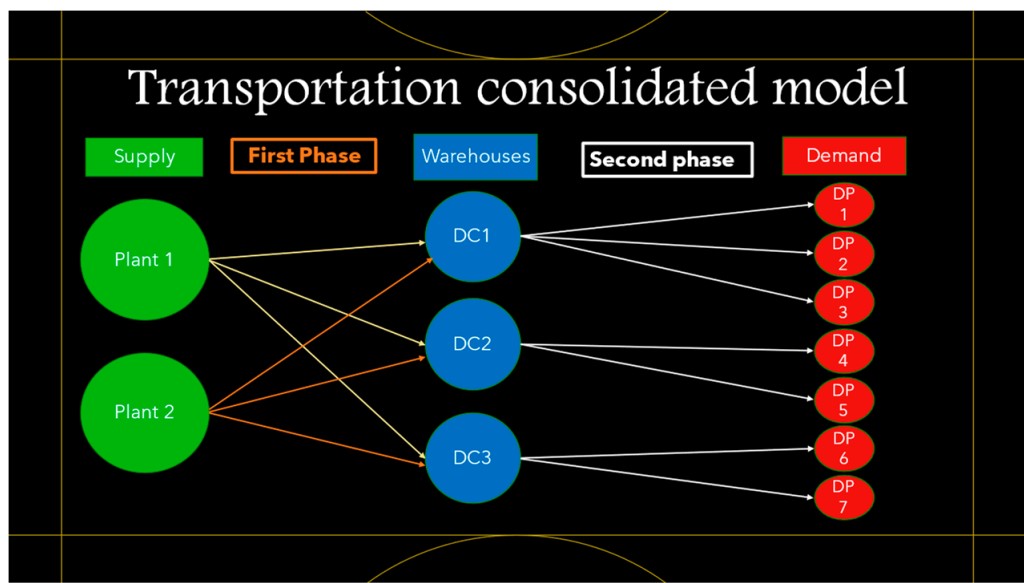

**Figure 6.** The transportation model with consolidation.

**Table 2.** Distances between plants and distribution centres (km).

| DC | Plant 1 | Plant 2 |
|---|---|---|
| DC 1 | 125 | 49.6 |
| DC 2 | 79 | 89.5 |
| DC 3 | 106 | 74.3 |

**Table 3.** Distances between distribution centres and demand points (km).

| DC | DP1 | DP2 | DP3 | DP4 | DP5 | DP6 | DP7 |
|---|---|---|---|---|---|---|---|
| DC 1 | 177 | 165 | 165 | - | - | - | - |
| DC 2 | - | - | - | 103 | 95.1 | - | - |
| DC 3 | - | - | - | - | - | 124 | 119 |

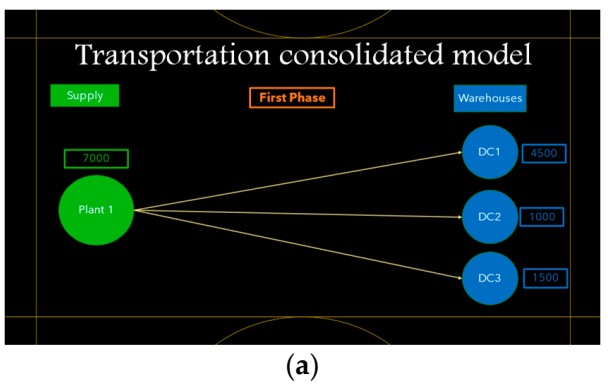
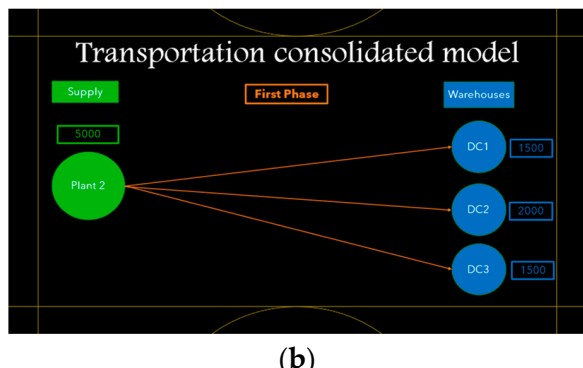

(**a**)　　　　　　　　　　　　　　　　　(**b**)

**Figure 7.** Phase 1 of the transportation model with consolidation for: (**a**) Plant 1, (**b**) Plant 2.

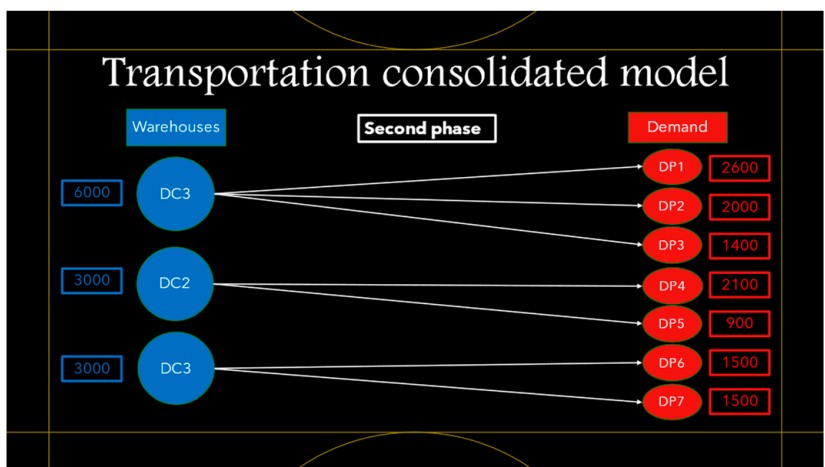

**Figure 8.** Phase 2 of the transportation model with consolidation.

Given that the weekly supply capacities for Plant 1 and Plant 2 are 7000 and 5000 cartons, respectively, phase 1 is modelled as follows:

Minimise $0.37116\,X_{11} + 0.23463\,X_{12} + 0.31482X_{13} + 0.147312X_{21} + 0.265815\,X_{22} + 0.220671\,X_{23}$

Subject to

$$X_{11} + X_{12} + X_{13} \leq 7000$$
$$X_{21} + X_{22} + X_{23} \leq 5000$$
$$X_{11} + X_{21} = 8000$$
$$X_{12} + X_{22} = 2000$$
$$X_{13} + X_{23} = 2000$$
$$X_{ij} \geq 0 \quad (\text{non} - \text{negativity constraint})$$

Phase 2 is modelled as follows:

Minimise $1.0681\,X_{11} + 0.89005\,X_{12} + 0.953625\,X_{13} + 1.05958\,X_{24} + 0.7582X_{25} + 0.76828X_{36} + 0.75343X_{37}$

Subject to

$$X_{11} + X_{12} + X_{13} \leq 6000$$
$$X_{24} + X_{25} \leq 3000 \quad X_{36} + X_{37} \leq 3000$$
$$X_{ij} \geq 0 \quad (\text{non} - \text{negativity constraint})$$

Excel Solver was used to solve these models, the total cost for the first phase of the consolidated model is 1823.68. In the second phase, the origins are the distribution centres and the destinations are seven branches of a large hypermarket chain, the total cost for

the second phase of the consolidated model is EGP 4135.5. Hence, the total cost for the consolidated model is EGP 5959.18 per week.

### 4.2. The Non-Consolidated Model

The non-consolidated model is the currently used model in the studied supply chain, hence there is no distribution centre in this model as shown in Figure 9. In this model, transportation costs are calculated directly for each route by using Equation (6). The total cost for this model is EGP 9962.95 per week. Figure 10 gives the comparison between the total costs of the consolidated model and the non-consolidated model. It was found that using the consolidated model has led to a 40% reduction in costs. In addition, as the fuel consumption is reduced in the consolidation model, less emissions are produced leading to a more sustainable logistics model. Hence, the findings validate the significance of the consolidation model as it does not only help in reducing the costs, but also it is a more environmentally friendly method, which is considered one of the most important aspects in logistics management. Moreover, it reduces the $CO_2$ and other emissions produced from the heavy-duty trucks. The results of this study agree with recent studies from the literature, and gives an opportunity to companies in Egypt to adopt shipment consolidation in their distribution strategy [31,33].

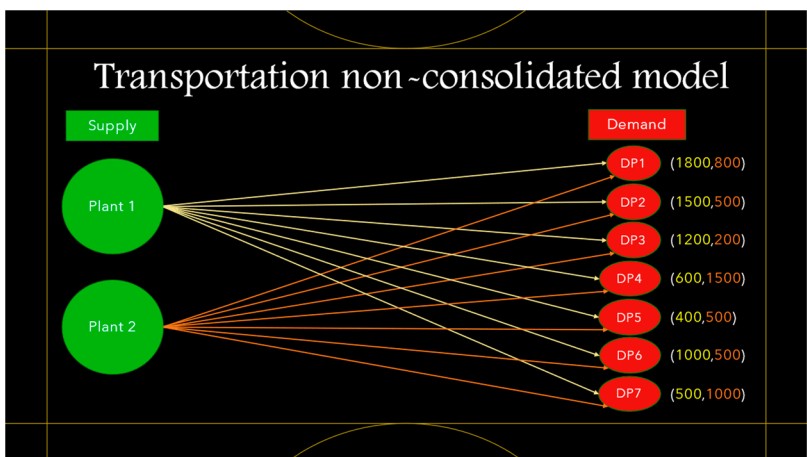

**Figure 9.** The transportation model with no consolidation.

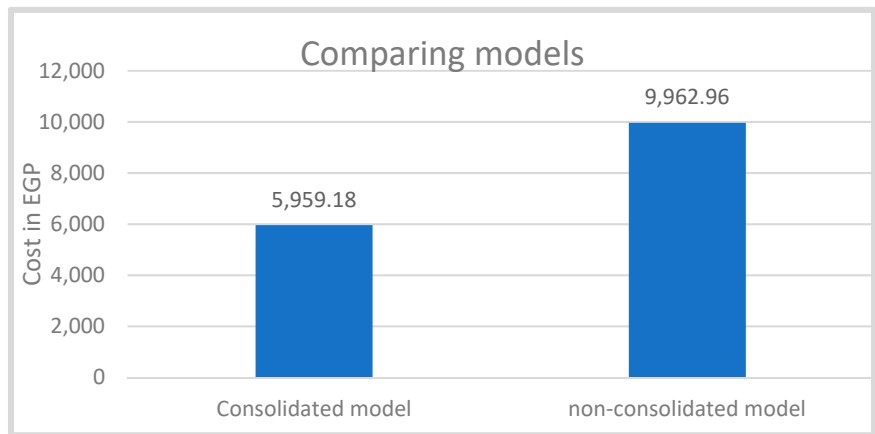

**Figure 10.** The total transportation costs for the two models.

The process of implementing shipment consolidation has strong implications for policy makers and top management. From cost- and service-level perspectives, shipment consolidation achieves better results; it should be included in both tactical and operational decision

levels. It is implied that significant savings can be achieved by performing shipment consolidation at a small subset of customer nodes, mid-route shipment consolidation can be used for carriers with non-clustered customer networks, and highly variable demand patterns.

**5. Conclusions**

Consolidation is not only a cost reduction method, but also it is a more efficient and environmentally friendly solution in logistics. Moreover, consolidation can solve traffic problems by decreasing the number of heavy trucks on the roads, and hence less maintenance will be required as heavy-duty trucks are responsible for the largest share of road usage. In this paper, a practical example of the benefits of consolidation was discussed. A case study from dairy supply chain was addressed, and two scenarios were studied; the first model used consolidation by using distribution centres as midpoints between the origins (milk plants) and the destinations (hypermarkets), while the second model uses direct shipment from the origins to the destinations. By comparing the two models, it was found that the consolidation method is an efficient, reliable, and environmentally friendly strategy as it could achieve cost reduction by more than 40%.

The cost of applying this model is relatively cheap if compared to the used model, as the extra expenses will be the distribution centres. These can be well managed if the principles of choosing the locations of these distribution centres, such as the centre of gravity method, are applied in addition to proper site analysis. Limitations of this paper are the relatively low number of covered works in the literature. This is due to the novelty of the shipment consolidation concept, and consequently the body of literature on its applications is relatively small. Another limitation is working on a small-sized network; however, this is a good direction for future work by extending the supply chain network to include more echelons. The consolidation model can be applied to other industries and the best example that shows the importance of consolidation is international shipping, as the shipping costs are much higher and to ship LCL is considered a waste of money that can be solved by using consolidation. Similarly, the last three years have witnessed a recession in the world economy due to the COVID-19 virus, shipment consolidation played a lifesaving role in this crisis by combining multiple shipments, hence reducing the cargo ships around the world which helped in stopping the spread of the virus. The multi-product pickup and delivery model can be used to develop a distribution network between multiple suppliers and developing an optimised model that is designed to manage the distribution network. Other directions for future research include fleet mixing with heterogeneous vehicles, and using delivery time-windows. Additionally, more insights on the impact of shipment consolidation on different parts of the supply chain need to be measured.

**Author Contributions:** Conceptualization, N.A.M.; methodology, N.A.M. and O.E.; software, O.E.; validation, O.E.; formal analysis, N.A.M.; O.E., writing—original draft preparation; O.E., writing—reviewing and editing; N.A.M., visualization, O.E.; supervision, N.A.M. All authors have read and agreed to the published version of the manuscript.

**Funding:** This research received no external funding.

**Institutional Review Board Statement:** Not applicable.

**Informed Consent Statement:** Not applicable.

**Conflicts of Interest:** The authors declare no conflict of interest.

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
