# Peer review of "A Sustainable Two-Echelon Logistics Model with Shipment Consolidation"

_logistics, 2023_

Round 1

Reviewer 1 Report

The aim of the paper is to develop a model that optimizes shipment consolidation while reducing the environmental damage, minimize the costs, and enhancing the process. Analytical models that are time-cost efficient and environmentally friendly are reviewed to assess the current situation of using that approach and develop a structured model.

This study is interesting, but there are statements unclear need to clarify, such as,

1.      The research questions and objectives identified from the study must be in line with the gaps identified from the study, include this in introduction.

2.      The literature review process carried out seems to be inappropriate. Add recent paper from 2019 to 2022 and compare the observations from the study with existing literature and explain in detail how the study stands different. 

3.      please identify the contribution of the study.

4.      The managerial, practical and theoretic implications of the study need to be detailed further.

5.      The conclusion section here is the mere extension of the results and it needs to be comprehensive further.

6.      The limitations of the research and scope for future works need to be explained further.

Author Response

1. The research questions and objectives identified from the study must be in line with the gaps identified from the study, include this in introduction.

Thank you for your comment, research questions are included in section 2.3.

2. The literature review process carried out seems to be inappropriate. Add recent paper from 2019 to 2022 and compare the observations from the study with existing literature and explain in detail how the study stands different. 

Thank you for your comment, papers from 2019-2022 are included in the references and was added in the results section.

3. please identify the contribution of the study.

Contributions are added into section 1.

4. The managerial, practical and theoretic implications of the study need to be detailed further.

Thank you for your comment, implications are added into section 4.

5. The conclusion section here is the mere extension of the results and it needs to be comprehensive further.

Thank you for your comment, Conclusions sections was modified.

6. The limitations of the research and scope for future works need to be explained further.

Conclusions sections was modified to include future research directions..

Reviewer 2 Report

Overall, this paper is not well written and the research paper is slightly interesting. However, the following comments shall be addressed:

1) Title is not impressive

Abstract required before introduction section also required to write properly. It should be written in the following steps;

a)            Write the concept as per your paper title (2 lines minimum).

b)            Objective (2 lines)

c)            Tools (2 lines)

d)            Output (3 lines)

e)            Output with application in real life or in the industry (mandatory)

2) In the Introduction section,

    a) the background of this research domain is way sufficing and the justification for this research (add some proper implication fig./picture in the background)

    b) research gap, weakly presented.

    c) the proposal of this research does not seem to be clearly presented to address the research gap. Please revise the Introduction section accordingly.

    d) Literature review shall be comprehensive (rather than brief) to discuss the right breadth of knowledge and recent works in the area.

   e) Better to add more recent work in the contribution table.

   f) Author should provide the real case study with an appropriate diagram/picture to explain with details.

   g) Managerial implications would be useful.

3) The English writing should be improved. For academic writing, try to avoid using Maybe, and, but etc. to start a sentence. Try to write research articles based on 3rd party writing style, hence avoid using We, our, etc.

4) Add conclusion section, Output with application in real life or in the industry (mandatory) in the conclusion section.

Please focus on the application.

Author Response

Title is not impressive

Thank you for your comment, the title was modified.

Abstract required before introduction section also required to write properly. Thank you for your valuable suggestion, the abstract was re-written as per your guidance.

The background of this research domain is way sufficing and the justification for this research (add some proper implication fig./picture in the background)

Thank you for your comment, the introduction was updated based on your suggestions.

Research gap, weakly presented.

Thank you for your comment, research gap was added.

The proposal of this research does not seem to be clearly presented to address the research gap. Please revise the Introduction section accordingly.

 Section 2.3 was updated to reflect on this.

Literature review shall be comprehensive (rather than brief) to discuss the right breadth of knowledge and recent works in the area.

Thank you for your comment, recent works were added to the review.

Author should provide the real case study with an appropriate diagram/picture to explain with details.

Thank you for your comment, Figure 7 and Figure 8 depict the case study.

Managerial implications would be useful.

Thank you for your comment, results and conclusions sections were updated based on your suggestion.

The English writing should be improved. For academic writing, try to avoid using Maybe, and, but etc. to start a sentence. Try to write research articles based on 3rd party writing style, hence avoid using We, our, etc.

Thank you for your remarks, the whole manuscript was revised for English spelling, grammar and syntax.

Add conclusion section, Output with application in real life or in the industry (mandatory) in the conclusion section.

Thank you for your comments, conclusions section was revised and updated.

Reviewer 3 Report

1.        The structure of the paper is good.

2.        The objective and contribution could be written more clearly in the introduction section.

3.        The literature review could show the contribution of the paper.

4.        The literature analysis for this article should cover a minimum of 40-50 items.

5.        I propose to include articles from the MDPI publishing house in the literature analysis.

6.        The case study is trivial, low scientific level.

Author Response

  1. The structure of the paper is good. Thank you for your time and efforts in reviewing the paper.
  2. The objective and contribution could be written more clearly in the introduction section.Thank you for your comment, introduction was updated.
  3. The literature review could show the contribution of the paper. Thank you for your remark, section 2.3 was added to address your suggestion.
  4. The literature analysis for this article should cover a minimum of 40-50 items. Thank you for your suggestion, due to page limitation and the novelty of the addressed topic, it was not possible to cover such number, however, based on your valuable comment, this was added as limitations int he conclusions section.
  5. I propose to include articles from the MDPI publishing house in the literature analysis. Thank you for your suggestion, more recent papers from MDPI were added.
  6. The case study is trivial, low scientific level. Thank you for your comment, the case study was used to exhibit the impact of shipment consolidation, it was based on real case study from the dairy industry in Egypt. To address your comment, that was added as a limitation of the paper and future research direction to expand the size of the supply chain network.

Author Response

After reading the paper carefully, my recommendation is major revision before acceptance for publication. The authors have addressed an original question, with smart experiments. This work provides an advance towards the current knowledge, clearly highlighted in the abstract. Taking the growing importance of logistics in the world economy and the very complex challenges in such hubs, the topic is up-to-date indeed. I realized that it is a very important study for the logistics and transportation field. 

Thank you very much for your time and efforts in reviewing our manuscript.
Therefore, I think there is an overall benefit to publish this work, after some minor formatting mistakes or typos, and a language check.

The whole manuscript was revised for English language, grammar, and syntax.

In my opinion, this paper is very interesting and deserves to attract a wide readership, beyond the limits of the journal's readership. The subject addressed in this paper is relevant. The study has been correctly designed, and is technically sound. HOWEVER, it’s a must that the authors improve the abstract that highlights the problem, background, methods, and also, main results and important contributions.

Thank you for your comment, the abstract was updated based on your suggestions.

In my opinion, the analyses of the results are convincing. HOWEVER, I would like to see this work revamped with further managerial implications. More analysis and discussion in terms of practical implications should be implemented. How are the model and the results beneficial to businesses and practitioners in the field of logistics and transportation. 

Thank you for your comment, results and conclusions were updated based on your suggestions.

The introduction highlights the context, and research goals and practical theoretical problems of the field as well. HOWEVER, points on sustainability are insufficient to me. I look forward to seeing more analysis on the importance of sustainable practices in the problem.

Thank you for your comment, the introduction was updated based on your suggestions.  

HOWEVER, the literature review is insufficient in my opinion. I suggest writing in an analyticalcritical-comparative style, processing the main international literature sources both from the practical and theoretical points of view. With 27 research references, it is suggested that the authors discuss some recent studies to
strengthen the work. Also, make a comparison between your model and other significant studies in the existing literature to underpin this paper’s novelty. For example, I would recommend these papers, which are among the latest works.

Thank you for your comment, the literature review was extended and the excellent papers you suggested were included.

Round 2

Reviewer 1 Report

THIS REVISED MANUSCRIPT HAS ADDRESSED MY CONCERN.

Author Response

Thank you for your efforts in reviewing our manuscript.

Reviewer 2 Report

Accept 

Author Response

Thank you for your time and efforts in reviewing our manuscript.

Reviewer 3 Report

The manuscript is revised according to my comments.

Author Response

(The authors gave the same response as above.)

Reviewer 4 Report

Thank you for your revision. The paper is now improved and can be significant for readers in the field. Thus, I accept this for publication, with minor revision on language style and appropriate academic writing.

Author Response

Thank you for your time and efforts, the whole paper has been revised for language and writing.